# Evidential Retriever: Uncertainty-Aware Medical Image Retrieval

**Sai Susmitha Arvapalli**                                    SUSMITHA@CSE.IITK.AC.IN
*IIT Kanpur, India*

**Vinay P. Namboodiri**                                        VPN22@BATH.AC.UK
*University of Bath, UK*

## Abstract

Medical image retrieval systems could play a vital role in clinical decision support by enabling physicians to find visually and semantically similar cases from large medical databases. However, deep learning-based retrieval models often overlook uncertainty in their predictions. To address this, we propose the Evidential Retriever, a novel architecture that combines evidential deep learning principles with transformer-based image representations to achieve more accurate and calibrated retrieval. Built upon a Swin Transformer backbone, our model features a dual-headed design: a retrieval head that performs metric learning for robust image embeddings, and an evidential head that models predictive uncertainty. We use a unified dual-loss, combining a regularized contrastive loss with an evidential loss. Experiments on three diverse medical imaging datasets: ISIC17, COVID-QU-Ex, and KVASIR - demonstrate that our method outperforms state-of-the-art retrieval models in retrieval accuracy and uncertainty estimation.

**Keywords:** Medical Image Retrieval, Evidential deep learning, Uncertainty Estimation.

## 1. Introduction

Content-Based Medical Image Retrieval (CBMIR) systems aim to retrieve clinically relevant and visually similar images from large-scale archives in response to a query image. This technology has potential to aid in clinical decision support, case-based reasoning, medical education, and differential diagnosis (Dubey, 2021; Choe et al., 2022; Manna et al., 2024). With the success of deep learning, modern CBMIR has shifted from handcrafted features to deep embeddings learned by Convolutional Neural Networks (CNNs)(Shetty et al., 2023; Hu et al., 2022) and, more recently, Vision Transformers (Arvapalli and Namboodiri, 2024; Trinh and Nguyen, 2021; Thakrar et al., 2023). These models learn powerful, low-dimensional representations that capture complex semantic content, leading to significant improvements in retrieval accuracy.

Despite this progress, a critical gap remains: reliability. Most deep retrieval models are deterministic. They minimize a prediction or metric loss, but the model is ignorant of its own confidence (Sensoy et al., 2018). When presented with a query, it will retrieve the "closest" matches from its embedding space, even if the match is ambiguous, poorly acquired, or from a completely unrelated domain (out-of-distribution). In a critical medical environment, this is a limitation. A model that retrieves an incorrect case can mislead a clinician.

Hence, there is an emergent need for retrieval systems that are not only accurate but also

uncertainty-aware (Cai et al., 2025). This can be solved to some extent using Bayesian Neural Networks (BNNs). However, BNNs often introduce significant computational overhead. Moreover, their uncertainty may not be well calibrated. A promising and more efficient alternative is Evidential Deep Learning (EDL) (Sensoy et al., 2018; Ulmer et al., 2023). EDL approaches the uncertainty problem from a 'Theory of Evidence' perspective. Instead of producing a simple softmax probability (a point estimate), an evidential network is trained to output the parameters of a Dirichlet distribution. This distribution models uncertainty over the class probabilities, directly quantifying the model's confidence based on the "evidence" it has collected from the data. In this paper, we introduce the Evidential Retriever, an architecture that integrates uncertainty quantification into a transformer model for CB-MIR. Unlike prior work that used evidential learning for classification-derived embeddings (Dordevic and Kumar, 2024), our model is an end-to-end unified framework that simultaneously learns discriminative embeddings and their associated evidential uncertainty. Our model feeds a shared [CLS] token representation into two parallel heads: An *Embedding head* trained with a deep metric learning loss to produce a discriminative embedding for retrieval. An *Evidential head* trained with an evidential loss to predict the Dirichlet parameter. Our design allows the model to simultaneously optimize for feature discrimination (for accurate retrieval) and evidence-based calibration (for reliable uncertainty). To summarize, our key contributions are:

- We propose the Evidential Retriever, a novel dual-head architecture that concurrently learns discriminative embeddings for retrieval and evidential parameters for uncertainty quantification in a single, end-to-end model.

- We introduce a composite loss function that effectively balances a deep metric learning objective for the embedding head with an evidential loss for the evidential head.

- We obtain state-of-the-art retrieval performance on diverse public medical datasets: ISIC (skin lesions), COVID-QU-Ex (chest X-Ray), and Kvasir (gastrointestinal endoscopy). We demonstrate qualitatively and quantitatively that our model's uncertainty estimates are calibrated, and provides ability for error-based filtration.

## 2. Related Work

Deep metric learning forms the foundation of image retrieval systems, where contrastive (Arvapalli and Namboodiri, 2024; El-Nouby et al., 2021) and triplet losses (Hu et al., 2022) promote compact and well-separated embeddings. Vision Transformers (ViTs) (Dosovitskiy et al., 2020) have recently outperformed CNNs by modeling global dependencies and learning richer representations. In medical image retrieval, studies on COVID, Kvasir, and ISIC datasets (Tschandl et al., 2019; Shetty et al., 2023; Agrawal et al., 2022) have primarily relied on CNN-based architectures like ResNet, VGG, and DenseNet. Subsequent works introduced improved similarity measures such as relative difference-based similarity (RDBSM) (Ahmed et al., 2023) and opponent class adaptive margin (OCAM) loss (Öztürk et al., 2023), while ViT-based methods (El-Nouby et al., 2021; Trinh and Nguyen, 2021; Thakrar et al., 2023; Gupta et al., 2023; Manzari et al., 2023; Arvapalli and Namboodiri, 2025) capture global contextual relationships more effectively. However, these approaches cannot quantify predictive reliability, an essential aspect in safety-critical domains like healthcare.

Uncertainty in deep learning is categorized into aleatoric (data-dependent) and epistemic (model) uncertainty. Gal and Ghahramani (Gal and Ghahramani, 2016) demonstrated that dropout applied at test time approximates Bayesian inference. (Lakshminarayanan et al., 2017) proposed Deep Ensembles, that trains multiple networks independently and combines their predictions. Other studies (Caldeira and Nord, 2020) have compared Bayesian methods, variational inference, and ensemble-based approaches for deep uncertainty quantification. Methods like Probabilistic Face Embeddings (PFE) (Shi and Jain, 2019) introduced amortized inference to quantify uncertainty by using auxiliary network heads. Subsequently, the Bayesian Triplet Loss (BTL) (Warburg et al., 2021) adopted this model structure to predict uncertainty but enforced constraints using a novel, analytically derived Bayesian triplet loss objective. BaNN's, Monte Carlo dropout, and deep ensembles are widely used but computationally expensive and require multiple forward passes or models. We use these methods as baselines. Note, our method estimates uncertainty in a single forward pass.

Evidential Deep Learning (EDL) is a principled framework for quantifying predictive uncertainty by interpreting network outputs as evidence distributions-Dirichlet for classification and Normal–Inverse-Gamma (NIG) for regression. Early works introduced evidential classifiers that decompose uncertainty without sampling (Sensoy et al., 2018), later extended to regression for single-pass uncertainty estimation (Amini et al., 2020). Prior Networks (Malinin and Gales, 2018) advanced distributional uncertainty modeling and OOD detection, influencing many calibration-oriented variants. Subsequent research explored pixel-level uncertainty in segmentation (Li et al., 2023), and large-scale comparisons (Schreck et al., 2023) showed that evidential models can approach ensemble-level uncertainty with lower computational cost. A recent survey by (Gao et al., 2024) offers a unified overview of EDL methods and challenges. Despite their adoption in classification and segmentation, evidential methods remain largely unexplored in image retrieval. The closest work (Dordevic and Kumar, 2024) relies on evidential classification foundations, i.e. they use classification-trained CLS token as an embedding. One of their other proposed approach is a Naive strategy to incorporate metric learning would require two networks - one for retrieval and one for uncertainty. Our method uses a single unified network. To the best of our knowledge, this is the first work to use evidential learning for medical image retrieval.

## 3. Method

Our approach, termed the **Evidential Retriever**, unifies evidential learning with deep metric learning to produce uncertainty-aware image embeddings.

### 3.1. Architecture

The proposed model is built on a transformer-based visual backbone, specifically the **Swin Transformer**, specifically swin_small has been chosen for its strong long-range and hierarchical feature capabilities (refer ablation study 4.1). To further enhance discriminative capability, we incorporate a cross-batch memory (Wang and Isola, 2020) that utilizes past embeddings as hard negatives, decoupling negative mining from batch size without extra computational cost. Given an input image $\mathbf{x}$, the Swin Transformer encodes it into a rich latent feature representation $\mathbf{h} \in \mathbb{R}^d$ using shifted-window attention blocks and patch-merging layers that preserve local and global contextual cues.

On top of this backbone, two lightweight task-specific heads are added:

- **Embedding Head:** A shallow two-layer feed-forward network consisting of a fully connected layer followed by a ReLU activation projects the pooled Swin features into a low-dimensional embedding space ($\mathbb{R}^{256}$). The embeddings are then $L_2$-normalized to ensure consistent magnitude across samples, which stabilizes contrastive learning.

- **Evidential Head:** A linear layer maps the same Swin features to class evidence logits, which are passed through a non-negative activation function (*Softplus*) to produce the evidence vector $\mathbf{e} = [e_1, \ldots, e_K]$. The corresponding Dirichlet parameters are computed as $\alpha_k = e_k + 1$, ensuring positivity. These parameters define a Dirichlet distribution that quantifies both belief and epistemic uncertainty.

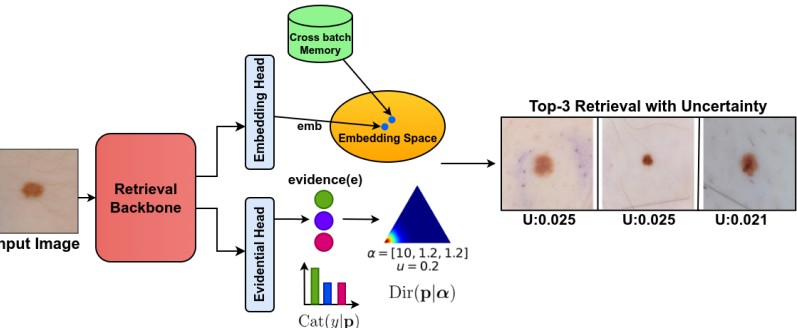

Figure 1: Architecture of the proposed evidential retrieval model. Producing $L_2$-normalized descriptors for contrastive learning with cross-batch memory from the embedding head and an evidential head that generates non-negative evidence converted into parameters $\boldsymbol{\alpha}$ of a Dirichlet Distribution ($\text{Dir}(\mathbf{p}|\boldsymbol{\alpha})$) over the class probability simplex, effectively quantifying the model's categorical belief ($\text{Cat}(\mathbf{y}|\mathbf{p})$) and its epistemic uncertainty ($u$). The final output shown is Top-3 Retrieval along with their respective Uncertainty($u$).

The expected class probabilities and total uncertainty are computed as:

$$\hat{p}_k = \frac{\alpha_k}{S}, \qquad u = \frac{K}{S}, \qquad S = \sum_{i=1}^{K} \alpha_i,$$

where $S$ denotes the total evidence strength. Higher $S$ indicates confident and reliable representations, while lower $S$ captures ambiguity or lack of evidence.

### 3.2. Loss Formulation

The network is optimized with a joint objective that balances retrieval performance and evidential calibration:

$$L_{\text{total}} = L_{\text{contr}} + \lambda_{\text{reg}} L_{\text{KoLeo}} + L_{\text{evid\_fit}},$$

**Contrastive Loss** ($L_{\text{contr}}$) ensures embeddings of semantically similar images are close in the learned space, while dissimilar ones are pushed apart, improving discriminative retrieval. It is defined as:

$$L_{\text{contr}} = \frac{1}{N} \sum_i \left[ \sum_{j:y_i=y_j} (1 - z_i^\top z_j) + \sum_{j:y_i \neq y_j} \max(0, z_i^\top z_j - \beta) \right] \tag{1}$$

where $z_i^\top z_j$ denotes the cosine similarity between $L_2$-normalized embeddings, $\beta$ is a margin controlling hard negatives, and $N$ is the number of samples in a batch. **KoLeo Regularization** ($L_{\text{KoLeo}}$) promotes geometric uniformity in the embedding space, preventing feature collapse. The regularization term is weighted by a coefficient $\lambda_{\text{reg}}$ that controls its contribution to the total loss.

$$L_{\text{KoLeo}} = -\frac{1}{N}\sum_{i=1}^{N}\log(\rho_i) \tag{2}$$

where $\rho_i = \min_{j\neq i}\|z_i - z_j\|$ is the distance between $z_i$ and its nearest neighbor. This term encourages embeddings to spread uniformly over the hypersphere, improving generalisation and retrieval robustness.

**Evidential Fit Loss** ($L_{\text{evid\_fit}}$) aligns the Dirichlet mean $\hat{p}_k$ with the one-hot label $y_k$, penalizing variance to encourage confident predictions when evidence is strong:

$$L_{\text{evid\_fit}} = \sum_k (y_k - \hat{p}_k)^2 + \frac{\alpha_k(S - \alpha_k)}{S^2(S+1)}. \tag{3}$$

### 3.3. Uncertainty-Aware Retrieval(Inference)

During inference, the retrieval embedding $\mathbf{z}$, obtained from the normalized Embedding Head, is used for similarity search via cosine similarity against the gallery. Separately, the Dirichlet-based uncertainty $u$ is calculated from the Dirichlet parameters ($\boldsymbol{\alpha}$) predicted by the Evidential Head, providing an interpretable, per-image measure of reliability. Images with higher uncertainty (low evidence) are flagged as ambiguous or out-of-distribution, while low-uncertainty samples indicate reliable matches. This unified formulation enables a single deterministic model to perform both high-accuracy feature retrieval and robust epistemic uncertainty estimation.

## 4. Results and Discussion

*Datasets:* We evaluate our proposed Evidential Retriever on three diverse medical imaging datasets covering different modalities. The ISIC Skin Lesion Dataset (Codella et al., 2018) includes 2,750 dermoscopic images of benign nevi, seborrheic keratosis, and melanoma, representing a fine-grained classification and retrieval task. The COVID-QU-Ex Dataset (Tahir et al., 2022) comprises 33,920 chest X-ray (CXR) images curated by researchers at Qatar University, categorized into 11,956 COVID-19 cases, 11,263 non-COVID infections (viral or bacterial pneumonia), and 10,701 normal cases. The dataset additionally provides ground-truth lung segmentation masks, enabling precise lung isolation for advanced retrieval tasks and making it the largest publicly available CXR dataset with lung masks. The Kvasir-V2 Dataset (Pogorelov et al., 2017) contains 8,000 endoscopic images categorized into eight classes, including anatomical landmarks and pathological findings.

We conduct experiments on three medical image retrieval benchmarks: ISIC, COVID-QU, and Kvasir-V2 to evaluate both retrieval performance and uncertainty reliability. To rigorously assess Out-of-Distribution (OOD) detection, we define distinct OOD pairs for each in-distribution (ID) dataset: for ISIC (ID), we use COVID-QU-Ex as the OOD set;

for COVID-QU-Ex (ID), we use Kvasir as the OOD set; and for Kvasir (ID), we use ISIC as the OOD set.

*Baselines:* Our comparisons span a diverse set of *deterministic*, *probabilistic*, *Bayesian*, and *evidential classification* retrieval models. Deterministic baselines include MIR-ViT (Arvapalli and Namboodiri, 2024), X-MIR (Hu et al., 2022), and Context-MIR (Arvapalli and Namboodiri, 2025); note that Context-MIR results are omitted for Kvasir as the method relies on segmentation maps, which are unavailable for this dataset. Probabilistic approaches such as Probabilistic Face Embeddings (PFE) (Shi and Jain, 2019) and Bayesian Triplet Loss (BTL) (Warburg et al., 2021) perform amortized inference to estimate the mean and variance of latent embeddings; for PFE, we incorporate an additional uncertainty head composed of `linear--BN--ReLU--linear--BN` layers while freezing the backbone parameters. We also evaluate approximate Bayesian methods including MC Dropout (Gal and Ghahramani, 2016) and Deep Ensembles (Lakshminarayanan et al., 2017), which have been widely used for uncertainty-aware retrieval. Further, we include the evidential classification approach of (Dordevic and Kumar, 2024).

*Evaluation metrics:* Across all methods, we assess image retrieval performance using Recall@K[1, 5, 10], mean Average Precision (mAP), and mean Precision@K (mP@K[1, 5, 10]). To evaluate uncertainty calibration on in-distribution (ID) data, we report the Expected Calibration Error (ECE). For out-of-distribution (OOD) detection, we measure the Area Under Receiver Operator Curve (AUROC) and Area Under Precision-Recall Curve (AUPRC), which quantifies the separability and ranking quality of uncertainty scores. Together, these metrics provide a comprehensive evaluation of retrieval effectiveness and uncertainty reliability. Additionally, for our proposed Evidential Retriever, we adopt $\lambda_{\text{reg}} = 0.7$, selected based on the hyperparameter tuning analysis detailed in Appendix .6.

From the quantitative results in Table 1, the proposed evidential transformer consistently provides the best overall balance between retrieval quality and uncertainty reliability across all three datasets. On ISIC, it achieves the highest mAP (73.65) and the strongest mP@K, while maintaining Recall@1 competitive with the strongest deterministic and probabilistic baselines; importantly, it delivers substantially superior OOD detection with AUROC 0.91 and AUPRC 0.99, while keeping ECE lower than most Bayesian baselines. On COVID-QU-Ex, our model attains the best retrieval scores (mAP 94.98 and the highest mP@K values), surpassing even strong deterministic retrieval models and the evidential classification rival, while simultaneously yielding the best OOD performance (AUROC 0.97, AUPRC 0.90) and a low ECE, indicating both accurate and well-calibrated predictions. A similar trend is observed on Kvasir-V2, where our method again attains the highest mAP (91.99) and strong Recall@K, with AUROC 0.95 and AUPRC 0.95 that markedly exceed all competing uncertainty methods, including BTL, PFE, Deep Ensembles, and pure evidential classification. Notably, while evidential classification is often the strongest competitor in OOD metrics, it underperforms our joint evidential transformer in retrieval scores and exhibits higher ECE, indicating that collapsing evidential learning into a classification-only head is less effective.

The robustness of the quantified uncertainty is further analyzed through Out-of-Distribution (OOD) detection capabilities, visualized in the density plots in Fig. 2. As observed in the ISIC density analysis, our Evidential Retriever achieves a distinct and clean separation between In-Distribution (ID) and OOD uncertainty distributions, with OOD samples con-

Table 1: **Quantitative Results.** Comparison of retrieval performance (Recall, mAP, mP), Out-of-Distribution (OOD) detection, and In-Distribution (ID) reliability across three medical imaging datasets. We compare against deterministic baselines (MIR-ViT (Arvapalli and Namboodiri, 2024), X-MIR (Hu et al., 2022), Context-MIR (Arvapalli and Namboodiri, 2025)), probabilistic methods (MC Dropout (Gal and Ghahramani, 2016), BTL (Warburg et al., 2021), PFE (Shi and Jain, 2019)), Deep Ensembles (Lakshminarayanan et al., 2017), and the Evidential Classification baseline (Dordevic and Kumar, 2024).

| | | IMAGE RETRIEVAL | | | OOD | | ID |
|---|---|---|---|---|---|---|---|
| | Model | Recall@[1,5,10] ↑ | mAP ↑ | mP@[1,5,10] ↑ | AUROC ↑ | AUPRC ↑ | ECE ↓ |
| **ISIC 2017** | MIR-ViT | [75.67, 87.33, 90.00] | 70.90 | [75.67, 74.50, 74.30] | - | - | - |
| | X-MIR | [80.67, 92.00, 96.00] | 69.29 | [80.67, 82.35, 82.89] | - | - | - |
| | Context-MIR | [74.00, 91.33, 96.00] | 71.33 | [74.00, 73.33, 73.87] | - | - | - |
| | MC Dropout | [72.66, 93.83, 97.50] | 63.52 | [72.66, 68.10, 67.70] | 0.3915 | 0.8405 | 0.1411 |
| | BTL | [80.66, 92.00, 94.00] | 68.62 | [80.66, 74.00, 73.20] | 0.4611 | 0.8306 | 0.3468 |
| | PFE | [73.00, 93.00, 96.83] | 61.47 | [73.00, 68.83, 67.93] | 0.6825 | 0.8815 | 0.1562 |
| | Deep Ensembles | [71.67, 93.33, 98.33] | 70.54 | [71.67, 72.20, 72.45] | 0.3497 | 0.7458 | **0.0660** |
| | Evidential Classif. | [74.83, 91.83, 95.00] | 70.45 | [74.83, 74.07, 73.68] | 0.6859 | 0.9785 | 0.2213 |
| | **Evidential (Ours)** | [79.67, 89.17, 91.00] | **73.65** | [79.67, 77.33, 76.98] | **0.9075** | **0.9876** | 0.1492 |
| **COVID-QU-Ex** | MIR-ViT | [93.80, 97.72, 98.42] | 91.43 | [93.80, 93.41, 93.36] | - | - | - |
| | X-MIR | [92.71, 97.73, 98.39] | 91.86 | [92.71, 92.56, 92.38] | - | - | - |
| | Context-MIR | [93.95, 97.08, 97.75] | 92.49 | [93.95, 93.56, 93.52] | - | - | - |
| | MC Dropout | [92.56, 97.61, 98.52] | 86.19 | [92.56, 91.77, 91.52] | 0.4710 | 0.2341 | 0.0908 |
| | BTL | [93.37, 98.39, 99.07] | 81.73 | [93.37, 92.84, 92.54] | 0.6480 | 0.2995 | 0.3973 |
| | PFE | [93.84, 97.81, 98.43] | 88.18 | [93.84, 92.84, 92.49] | 0.7149 | 0.3450 | 0.0868 |
| | Deep Ensembles | [94.74, 98.38, 98.75] | 93.74 | [94.74, 94.60, 94.35] | 0.3233 | 0.1898 | 0.0698 |
| | Evidential Classif. | [93.17, 97.86, 98.71] | 89.76 | [93.17, 92.47, 92.22] | 0.8936 | 0.7877 | 0.0902 |
| | **Evidential (Ours)** | [95.67, 97.02, 97.48] | **94.98** | [95.67, 95.47, 95.44] | **0.9717** | **0.8959** | **0.0581** |
| **KVASIR** | MIR-ViT | [93.79, 97.42, 97.96] | 90.97 | [93.79, 93.56, 93.52] | - | - | - |
| | X-MIR | [90.92, 97.67, 98.75] | 88.68 | [90.92, 90.76, 90.72] | - | - | - |
| | Context-MIR | - | - | - | - | - | - |
| | MC Dropout | [91.54, 97.25, 98.16] | 89.23 | [91.54, 91.23, 91.20] | 0.7145 | 0.2805 | 0.0957 |
| | BTL | [92.87, 98.29, 98.83] | 88.20 | [92.87, 92.30, 92.29] | 0.7612 | 0.1055 | 0.3115 |
| | PFE | [91.87, 97.50, 98.37] | 89.02 | [91.87, 91.00, 90.72] | 0.3840 | 0.1544 | 0.1069 |
| | Deep Ensembles | [91.95, 97.08, 97.95] | 90.57 | [91.95, 92.32, 92.28] | 0.4253 | 0.1783 | 0.0654 |
| | Evidential Classif. | [92.29, 97.71, 98.50] | 89.96 | [92.25, 91.91, 91.70] | 0.9148 | 0.6325 | 0.0776 |
| | **Evidential (Ours)** | [93.08, 97.54, 98.33] | **91.99** | [93.08, 93.17, 93.43] | **0.9543** | **0.9517** | **0.0593** |

sistently pushed toward higher uncertainty scores. This separation is statistically corroborated by the highest AUROC and AUPRC scores across all datasets. In contrast, amortized inference methods such as PFE and BTL show significant overlap between ID and OOD distributions. This highlights a fundamental limitation of amortization: while effective at modeling aleatoric data noise within the training distribution, it struggles to extrapolate epistemic uncertainty for distinct, unseen distributions. While the Evidential Classification baseline also exhibits a similar but consistently weaker separation, our model demonstrates a more balanced distribution that supports both OOD detection and granular reliability assessment for retrieval candidates, a trend that is consistent across the COVID-QU-Ex and Kvasir datasets.

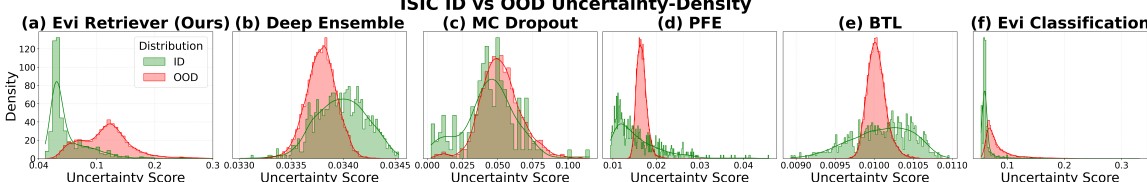

Figure 2: **Qualitative Safety Analysis (ID vs. OOD).** Density histograms of uncertainty scores for In-Distribution (green) ISIC samples and OOD as (red) COVID-QU-Ex samples across six retrieval baselines. Deep Ensembles, MC Dropout, and BTL (b,c,e) exhibit variance collapse, where the ID and OOD distributions either overlap heavily or degenerate into narrow spikes, resulting in unreliable OOD detection. PFE and Evi Classification (d, f) shows partial separation. **Our method** (a) achieves clear semantic separation: ID samples form a compact low-uncertainty mode, while OOD samples shift distinctly toward higher uncertainty, aligning with the superior AUROC.

The practical utility of this uncertainty is demonstrated by the sparsification curves in Fig. 3, which plot the improvement in retrieval accuracy (mAP@1) as the most uncertain samples are progressively filtered out. Across all three datasets we display the top-3 performing curves - corresponding to Evidential Retriever(ours), BTL, and Evidential Classification. The curves for the Evidential Retriever rise monotonically, confirming that the model reliably associates higher uncertainty with harder or erroneous observations. Crucially, our model distinguishes itself from the baselines in two key aspects: the starting point and the rate of improvement. The Evidential Retriever consistently begins at the highest base accuracy (0% filter rate) and maintains a steep, robust ascent in the critical low-rejection regime. In comparison, BTL shows a positive correlation but starts from a lower performance baseline and its practical utility is compromised by the very poor OOD detection performance observed in Table 1. More importantly, our model significantly outperforms the Evidential Classification baseline in this metric. The sparsification curves for the classification approach often start lower and exhibit less effective filtering (as seen in the flatter trajectory on COVID-QU-Ex), suggesting that while the Evidential classification captures global uncertainty, it is less correlated with specific retrieval errors than the uncertainty derived from our unified metric learning framework. These results show that our approach assigns the highest uncertainty to genuinely hard cases, making its uncertainty estimates actionable. For a visual examination of these flagged cases, we provide a detailed qualitative analysis of the high-uncertainty samples in Appendix .2 (ISIC), .3 (COVID-QU-Ex), and .4 (Kvasir) and retrieval analysis in .5.

## 4.1. Ablation Studies

*Effect of Backbone Choice:* To investigate the impact of architectural design on retrieval quality and uncertainty modeling, we evaluate four transformer backbones: ViT-Small, ViT-Base, Swin-Tiny, and Swin-Small - across all three datasets. Table 2 summarizes the retrieval performance in terms of R@K, mAP, mP@K, and calibration metrics. The hierarchical transformer backbones consistently outperform the ViT models, indicating that multi-scale feature aggregation and localized windowed attention are more effective for

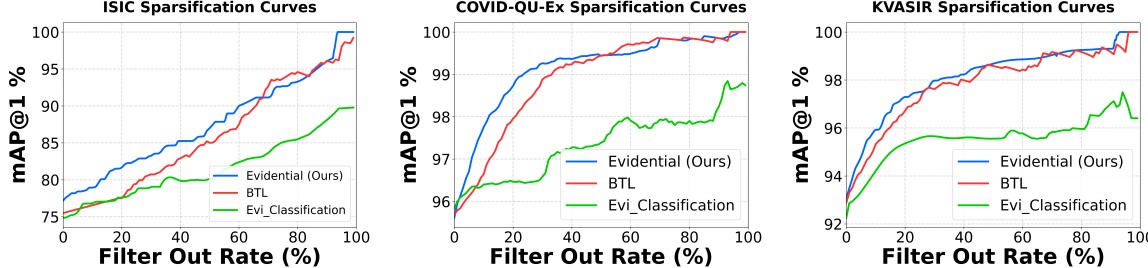

Figure 3: **Uncertainty Utility (Sparsification Curves).** These plots show the mAP@1 gain versus the Filter Out Rate (%) for three models across ISIC, COVID-QU-Ex, and KVASIR datasets. The monotonically increasing curves confirm that uncertainty correlates with observation difficulty. Our Evidential Retriever (Blue) achieves the highest mAP@1 gain, even at low Filter Out Rates, validating the superior calibration of its uncertainty scores for challenging samples.

Table 2: Backbone comparison across ISIC, Kvasir, and COVID-QU-Ex using Recall@K, mAP, mP@K, and ECE. Overall, **Swin Transformers** outperform ViT variants, with **Swin-Small** offering the best balance between high retrieval accuracy and reliable uncertainty calibration. Top scores are highlighted in bold.

| Dataset | Backbone | R@K | mAP | mP@K | ECE |
|---|---|---|---|---|---|
| ISIC | Swin-Small | [79.67, 89.17, 91.00] | **73.65** | [79.67, 77.33, 76.98] | 0.1492 |
| | Swin-Tiny | [76.17, 89.33, 92.67] | 72.05 | [76.17, 76.17, 75.68] | **0.1310** |
| | ViT-Base | [73.33, 88.33, 91.17] | 70.38 | [73.33, 73.57, 73.50] | 0.1775 |
| | ViT-Small | [75.17, 90.17, 94.00] | 71.40 | [75.17, 75.33, 75.02] | 0.1522 |
| Kvasir | Swin-Small | [93.08, 97.54, 98.33] | **91.99** | [93.08, 93.17, 93.43] | **0.0593** |
| | Swin-Tiny | [93.08, 97.25, 97.92] | 90.44 | [93.08, 93.07, 93.11] | 0.0643 |
| | ViT-Base | [93.29, 96.25, 97.04] | 88.80 | [93.29, 92.18, 92.07] | 0.0743 |
| | ViT-Small | [93.33, 96.50, 97.71] | 89.51 | [93.33, 93.28, 93.09] | 0.1096 |
| COVID-QU-Ex | Swin-Small | [95.67, 97.02, 97.48] | **94.98** | [95.67, 95.47, 95.44] | **0.0581** |
| | Swin-Tiny | [95.60, 97.70, 98.14] | 93.93 | [95.60, 95.12, 94.99] | 0.0721 |
| | ViT-Base | [94.83, 97.69, 98.19] | 91.96 | [94.83, 94.44, 94.18] | 0.0599 |
| | ViT-Small | [94.80, 97.58, 98.03] | 92.72 | [94.80, 94.51, 94.35] | 0.0704 |

modeling fine-grained medical structures such as dermoscopic lesion borders and polyp textures. Between the two Swin variants, Swin-Small shows a consistent advantage over Swin-Tiny across datasets, reflecting the benefits of deeper stages and richer multi-scale representations. Considering the overall balance between retrieval accuracy, uncertainty calibration, and computational efficiency, Swin-Small emerges as the strongest backbone and is therefore selected as the backbone for our Evidential Retriever model.

*Ablation Study: Effect of Loss Functions:* To assess how different loss formulations affect retrieval quality and uncertainty modeling, we compare five configurations across all three medical imaging datasets as shown in Table 3. The evidential-only model ($L_{\text{evid\_fit}}$) outperforms the standard contrastive baseline ($L_{\text{contr}}$) on both the ISIC and Kvasir datasets,

Table 3: Ablation study of the three loss components - contrastive ($L_{\mathrm{contr}}$), KoLeo ($L_{\mathrm{KoLeo}}$), and evidential fit loss ($L_{\mathrm{evid\_fit}}$) on ISIC, COVID-QU-Ex, and Kvasir. Each component improves performance, and the full combination achieves the best mAP across all datasets, confirming their complementary contributions to retrieval quality.

| Loss Components | | | Datasets | | |
|---|---|---|---|---|---|
| $L_{\mathbf{contr}}$ | $L_{\mathbf{KoLeo}}$ | $L_{\mathbf{evid\_fit}}$ | **ISIC** | **COVID-QU-EX** | **Kvasir** |
| | | | $mAP$ | $mAP$ | $mAP$ |
| – | – | ✓ | 70.45 | 89.76 | 89.96 |
| ✓ | – | – | 68.20 | 91.43 | 89.41 |
| ✓ | – | ✓ | 69.67 | 93.22 | 90.34 |
| ✓ | ✓ | – | 71.33 | 92.49 | 90.97 |
| ✓ | ✓ | ✓ | **73.65** | **94.98** | **91.99** |

achieving an mAP of 70.45% and 89.96% respectively. This suggests that the evidential objective, while formulated for classification, imposes strong class-discriminative constraints that implicitly structure the shared feature space effectively for retrieval. However, on the COVID-QU-Ex dataset, the contrastive baseline proves superior, indicating that pairwise metric learning is still essential for certain data distributions. We observe that combining components leads to further improvements; for instance, adding evidential supervision to the contrastive loss ($L_{\mathrm{contr}} + L_{\mathrm{evid\_fit}}$) boosts performance on COVID-QU-Ex to 93.22%, while incorporating KoLeo regularization ($L_{\mathrm{contr}} + L_{\mathrm{KoLeo}}$) is particularly effective on ISIC, raising the mAP to 71.33% by mitigating feature collapse. Across all datasets, the best performance is consistently achieved by our unified loss $L_{\mathrm{total}}$, which combines regularized contrastive learning with evidential modeling. For a qualitative visual comparison of the embedding spaces learned by the evidential classification baseline and our Evidential Retriever, please refer to the t-SNE analysis in Appendix .1.

## 5. Conclusion

In this work, we introduce the Evidential Retriever, a framework that unifies discriminative representation learning with evidential uncertainty modeling for medical image retrieval. By extending the Dirichlet-based evidential formulation, our approach enables the model to generate feature representations that encode both semantic similarity and uncertainty. Through evaluation on three diverse medical imaging benchmarks: ISIC, COVID-QU-Ex, and Kvasir, we demonstrate that the Evidential Retriever achieves consistently superior retrieval performance while offering meaningful uncertainty estimates that strongly correlate with embedding quality. Our experiments confirm that the model distinguishes between in-distribution and out-of-distribution samples and provides uncertainty signals for error filtration.

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

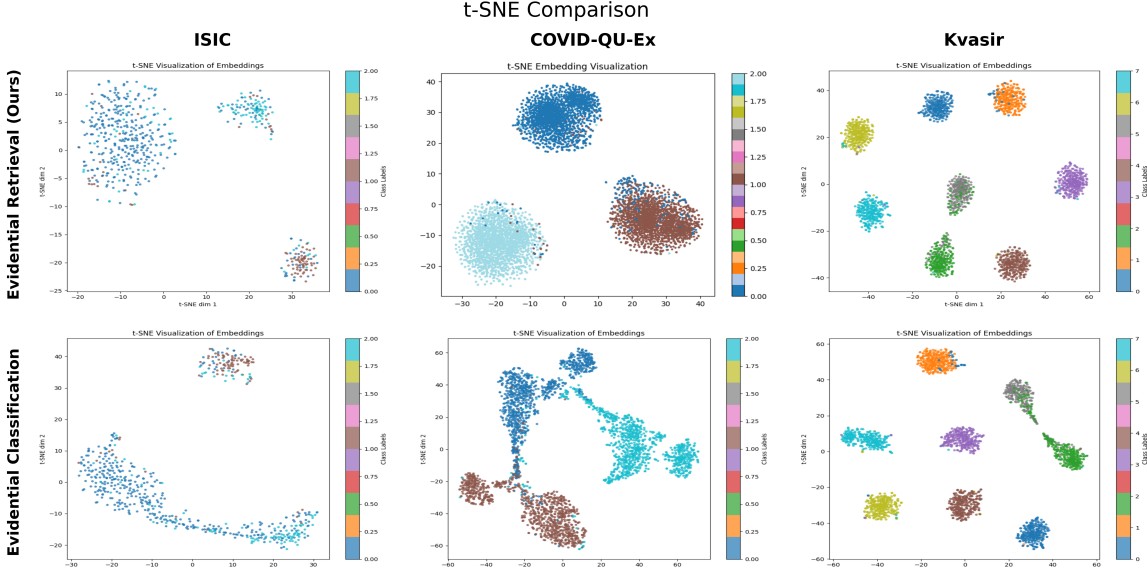

Figure 4: t-SNE visualization of test set embeddings. The top row displays the feature space learned by our proposed **Evidential Retrieval** framework, while the bottom row shows the **Evidential Classification** baseline. Our method demonstrates significantly improved intra-class compactness and inter-class separability, particularly visible in the ISIC and COVID-QU-Ex datasets.

## .1. Qualitative Analysis of Feature Embeddings

To further validate the discriminative capability of our proposed Evidential Retriever, we visualize the learned latent feature spaces using t-Distributed Stochastic Neighbour Embedding (t-SNE). Figure 4 presents a side-by-side comparison of the test set embeddings generated by our method versus the Evidential Classification baseline across the ISIC, COVID-QU-Ex, and Kvasir datasets. As observed in the visualizations, the Evidential Retriever produces significantly more structured feature representations characterized by improved intra-class compactness and inter-class separability. This is particularly evident in the COVID-QU-Ex dataset, where the baseline classification model yields dispersed and elongated clusters, whereas our method condenses these into distinct, spherical distributions. Similarly, for the ISIC dataset, our approach enforces a clearer margin between different classes compared to the baseline, which exhibits blurred boundaries. We attribute this structural improvement to our unified dual-head framework, which seamlessly integrates uncertainty quantification into the metric learning objective. By jointly optimizing the evidential head and the embedding head, the model effectively leverages uncertainty estimates to regularize the latent space, penalizing ambiguous overlap and encouraging the formation of high-density, class-specific manifolds. This ensures that the learned metric space is not only discriminative but also calibrated, directly supporting the quantitative performance gains reported in Table 1.

**Low Uncertainty Gallery (Top 15)**

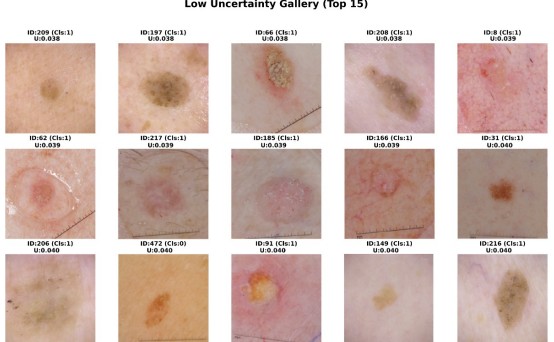

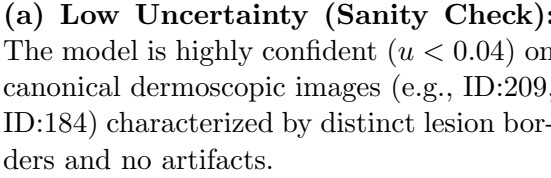

**High Uncertainty Gallery (Top 15)**

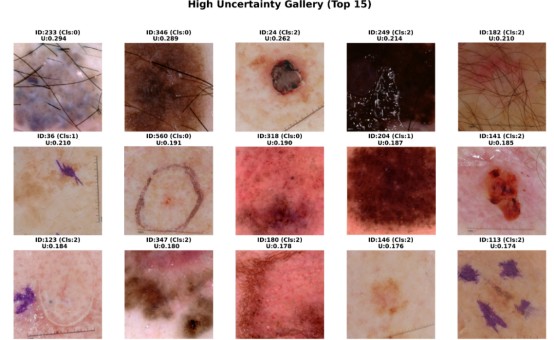

**(a) Low Uncertainty (Sanity Check):** The model is highly confident ($u < 0.04$) on canonical dermoscopic images (e.g., ID:209, ID:184) characterized by distinct lesion borders and no artifacts.

**(b) High Uncertainty (Safety Mechanism):** The model correctly flags reliability risks including **Occlusion** (e.g., ID:233: Dense Hair), **Foreign Objects** (e.g., ID:36: Sutures), and **Acquisition Artifacts** (e.g., ID:249: Gel/Bubbles).

Figure 5: **Qualitative Analysis on ISIC Dataset.** The uncertainty score acts as a quality filter, distinguishing between clear diagnostic samples and inputs degraded by occlusion or synthetic artifacts.

## .2. Qualitative Analysis of Uncertainty (ISIC)

We validated the clinical reliability of the Evidential Retriever by analyzing its uncertainty estimation on the ISIC skin lesion dataset. As shown in the low-uncertainty gallery (Fig. 5, a), the model assigns minimal uncertainty ($u < 0.04$) to canonical dermoscopic samples - such as ID:209 and ID:184 - which represent the ideal diagnostic scenario characterized by high-contrast lesions, distinct borders, and a complete absence of obstruction; this confirms the model's robustness on the clean, high-density regions of the training distribution. Conversely, the high-uncertainty gallery (Fig. 5, b) reveals a robust safety mechanism that detects image quality degradation and external artifacts. Specifically, the model flags **severe occlusion**, where dense hair blocks the lesion (e.g., ID:233), **foreign object interference**, such as purple surgical sutures (ID:36) or ink annotations (ID:113), and **acquisition artifacts**, where immersion fluid bubbles create discordant textures (ID:249). The t-SNE visualization (Fig. 8(a)) corroborates these findings, showing that the most uncertain samples (marked with Red Stars) cluster in sparse regions or at the manifold periphery, confirming that the model correctly isolates these anomalous inputs from the reliable distribution.

## .3. Qualitative Analysis of Uncertainty (COVID-QU-Ex)

To demonstrate the model's reliability on the COVID-QU-Ex dataset, we examined images with the lowest and highest uncertainty scores. As shown in the low-uncertainty gallery (Fig. 6, a), the model is most confident ($u \approx 0.05$) on clear, standard X-rays (e.g. ID:4811) that look exactly like the typical training data. In contrast, the high-uncertainty gallery (Fig. 6, b) shows that the model correctly flags "odd" or difficult images as unreliable. This

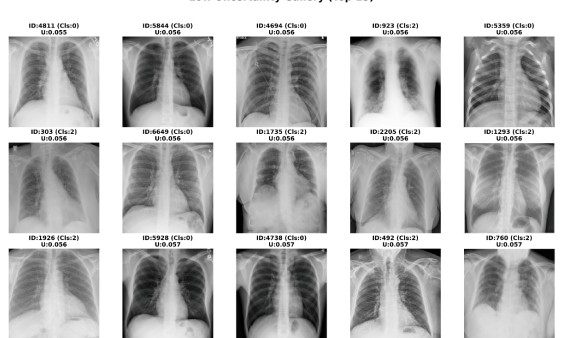
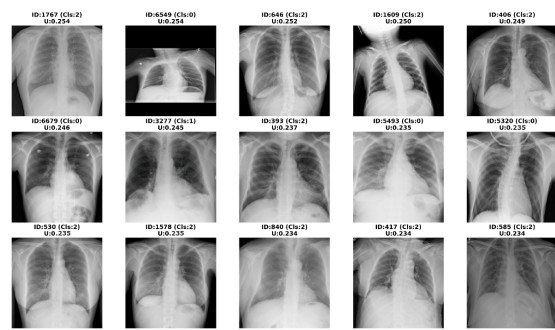

**(a) Low Uncertainty (Sanity Check):** The model is most confident ($u \approx 0.05$) on high-quality, standard X-rays (e.g., ID:4811) that are clear, upright, and free of artifacts.

**(b) High Uncertainty (Safety Mechanism):** The model acts as a safety net by correctly flagging unreliable images. This includes **Technical Errors** (e.g., ID:1609: Pediatric, ID:6549: Rotated Image) and **Difficult Medical Cases** (e.g., ID:5320: Severe Spine Curvature, ID:393: Obscured Lungs).

Figure 6: **Qualitative Analysis on COVID-QU-Ex Dataset.** The uncertainty score reliably distinguishes between standard, clear images and complex or erroneous inputs that require human review.

includes technical errors-such as scans of **children** (ID:1609), **rotated images** (ID:6549), or wires blocking the view (ID:406)-as well as unusual medical cases like severe spinal curvature (ID:5320) or completely **obscured lungs** (ID:393). The t-SNE plot (Fig. 8 (b)) confirms this behavior, showing that these uncertain images are pushed to the edges of the data clusters, far from the standard examples.

### .4. Qualitative Analysis of Uncertainty (Kvasir)

To validate the clinical safety of the Evidential Retriever, we qualitatively analyze retrieval behaviors under varying levels of epistemic uncertainty. As shown in the low-uncertainty gallery (Fig. 7, a), the model demonstrates maximum confidence ($u \approx 0.08$) on canonical samples belonging to **Class 4 (Pylorus)**, where distinct anatomical landmarks facilitate robust retrieval. Conversely, the high-uncertainty gallery (Fig. 7, b) highlights the model's capacity to flag reliability risks. We observe two predominant sources of uncertainty: (1) **acquisition artifacts**, where samples containing non-biological features like green UI overlays (e.g., ID:1210) are treated as out-of-distribution, and (2) **semantic ambiguity**, where borderline cases (e.g., ID:118) lie on the decision boundary between **Normal Z-line** and **Esophagitis**. The corresponding t-SNE visualization (Fig. 8 (c)) corroborates these observations, showing that high-uncertainty samples cluster at class peripheries or in sparse manifold regions. This confirms that the evidential head effectively grounds uncertainty in the geometry of the embedding space.

**Low Uncertainty Gallery (Top 15)**

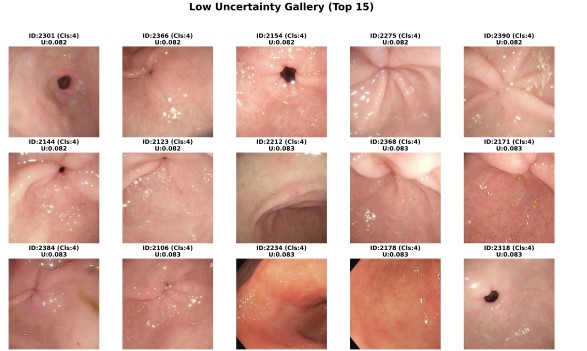

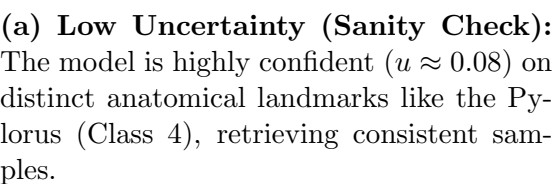

**High Uncertainty Gallery (Top 15)**

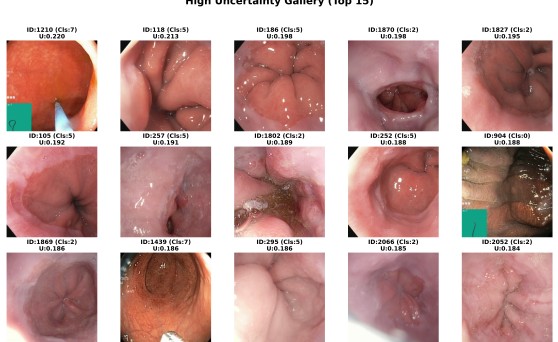

**(a) Low Uncertainty (Sanity Check):** The model is highly confident ($u \approx 0.08$) on distinct anatomical landmarks like the Pylorus (Class 4), retrieving consistent samples.

**(b) High Uncertainty (Safety Mechanism):** The model flags two types of reliability risks: (1) **OOD Artifacts** (e.g., ID:1210 with green UI overlay) and (2) **Medical Ambiguity** (e.g., ID:118), where the distinction between Normal Z-line and Esophagitis is visually subtle.

Figure 7: **Qualitative Analysis on Kvasir Dataset.** The uncertainty score serves as an effective reliability indicator, distinguishing between clear anatomical features and ambiguous or artifact-laden inputs.

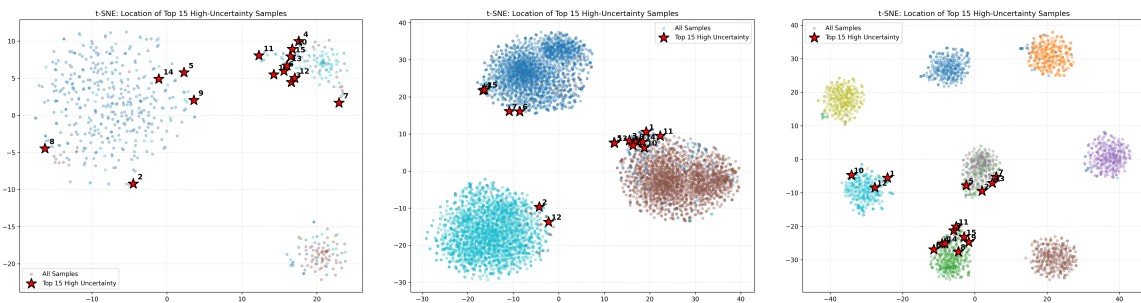

**(a) ISIC (Dermatology) (b) COVID-QU-Ex (X-Ray) (c) Kvasir (Endoscopy)**

Figure 8: **Geometric Interpretation of Uncertainty across Datasets.** t-SNE visualizations showing the embedding space for (a) ISIC, (b) COVID-QU-Ex, and (c) Kvasir. In all cases, the top 15 most uncertain samples (Red Stars) are not randomly distributed; they consistently cluster in sparse regions, at manifold edges (representing artifacts/OOD), or along ambiguous decision boundaries. This confirms that the evidential uncertainty is geometrically well-grounded.

### .5. Qualitative Analysis of Retrieval Safety

To validate the practical utility of the Evidential Retriever, we conducted a retrieval case study using the specific samples identified as the least and most uncertain in our previous global analysis in Sec. .2, .3, .4. By querying with these exact outliers, we directly observe

**ISIC Top-3 Retrievals**

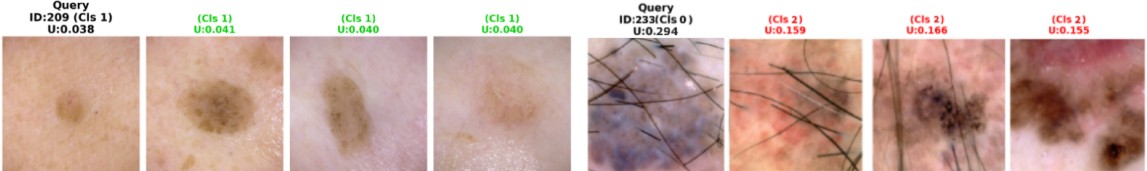

(a) **ISIC (Texture Bias):** Querying with a clear lesion (Left) yields accurate neighbors. Querying with a hair-occluded outlier (Right) triggers texture bias-retrieving irrelevant "hairy" images—but is safely flagged by high uncertainty ($u = 0.294$).

**COVID-QU-Ex Top-3 Retrievals**

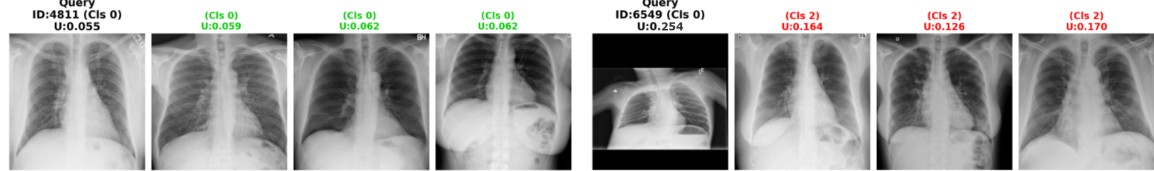

(b) **COVID-QU-Ex (Geometric Shift):** The canonical upright scan (Left) is retrieved robustly. The rotated outlier (Right) causes mixed-class retrieval due to domain shift but is correctly identified ($u = 0.254$) as unreliable.

**KVASIR Top-3 Retrievals**

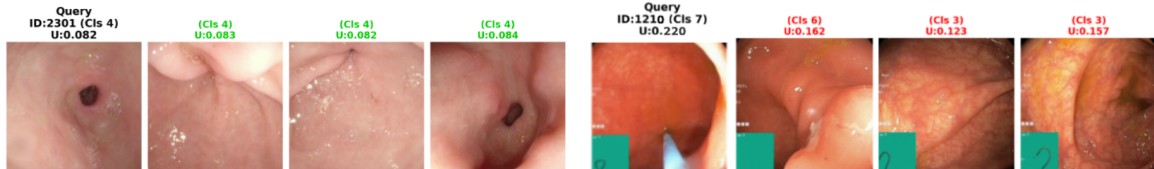

(c) **Kvasir (Artifacts):** Distinct anatomy (Left) succeeds. The green-artifact outlier (Right) retrieves spurious noise sharing the same artifact, which is effectively flagged ($u = 0.220$) to prevent misdiagnosis.

Figure 9: **Qualitative Analysis of Retrieval Safety.** We visualize the top-3 retrieved neighbors for the highest and lowest uncertainty samples identified in Sec. .2, .3, and .4. Retrieval correctness is color-coded (**Green**: Correct Class, **Red**: Incorrect Class). The figure is organized by dataset: **(a) ISIC** (Texture Bias), **(b) COVID-QU-Ex** (Geometric Shift), and **(c) Kvasir** (Artifacts). While "Ideal Success" queries (Left) yield consistently accurate (green) retrievals, "Safe Failure" queries (Right) demonstrate that the model retrieves incorrect (red) classes when inputs are degraded by occlusion, rotation, or artifacts; crucially, the high uncertainty assignment effectively prevents these from becoming silent failures.

the downstream consequences of data irregularities in Fig. 9. In the low-uncertainty scenarios (Left Column), the model consistently retrieves semantically relevant neighbors, con-

firming its robustness on canonical, high-quality data. Conversely, querying with the flagged high-uncertainty samples (Right Column) reveals distinct failure modes: hair-occluded skin lesions trigger texture bias (retrieving other hairy images), rotated chest X-rays cause geometric confusion, and artifact-laden endoscopy images lead to the retrieval of synthetic noise. Crucially, however, the high evidential uncertainty scores correctly identify these predictions as unreliable. This confirms the safety utility of our method: it effectively warns clinicians when retrieval results are driven by occlusion, distribution shifts, or artifacts, thereby preventing "silent failures" in the decision-making process.

## .6. Hyper parameter tuning - Effect of $\lambda_{\mathrm{reg}}$ in Metric Learning

To determine the optimal balance between standard metric learning and evidential regularization, we conduct a systematic hyperparameter sweep over the evidential weighting factor $\lambda_{\mathrm{reg}} \in \{0.0, 0.3, 0.7\}$ using the validation sets of ISIC, COVID-QU-Ex, and Kvasir-V2 as shown in Figure 10. For all experiments, we use the Swin-Small backbone and evaluate retrieval quality using Recall@K, mAP, and mP@K. Across all three datasets, $\lambda_{\mathrm{reg}} = 0.0$ (i.e., no regularization) consistently results in the weakest retrieval performance, indicating that regularization plays a crucial role in stabilizing the embedding space. The ISIC dataset achieves its highest validation performance at $\lambda_{\mathrm{reg}} = 0.3$, followed by $\lambda_{\mathrm{reg}} = 0.7$. In both COVID-QU-Ex and Kvasir-V2 obtain their strongest Recall@K, mAP, and mP@K metrics at $\lambda_{\mathrm{reg}} = 0.7$, demonstrating that stronger evidential regularization improves robustness and calibration for datasets with more homogeneous structural patterns. Since $\lambda_{\mathrm{reg}} = 0.7$ performs best on two out of the three datasets and remains competitive on ISIC, we adopt $\lambda_{\mathrm{reg}} = 0.7$ as the unified hyperparameter for all subsequent experiments, offering the best trade-off between retrieval accuracy and uncertainty-aware representation learning.

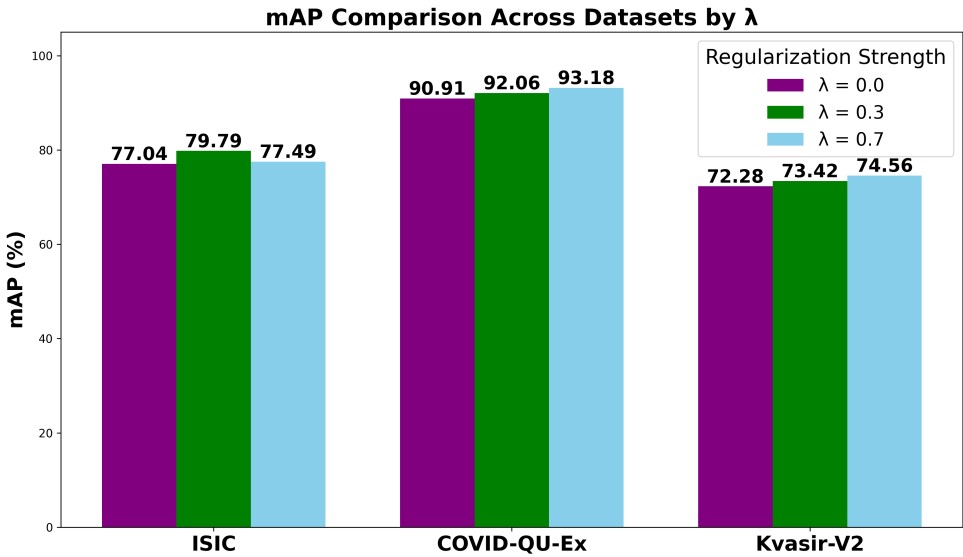

Figure 10: Comparison of mAP Scores Across Datasets for Different Regularization Strengths $\lambda_{\mathrm{reg}}$.

### .7. Implementation Details

We adopt the training recipe outlined in (Arvapalli and Namboodiri, 2024). All models are optimized using the AdamW optimizer with a learning rate of $3 \times 10^{-5}$ and a weight decay of $5 \times 10^{-4}$ for 10,000 iterations. For the contrastive objective, the margin is set to $\beta = 0.5$. To analyze the effect of regularization, we experiment with weighting factors $\lambda_{\text{reg}} \in \{0.0, 0.3, 0.7\}$. Standard data augmentation techniques are applied during training, including resizing images to $256 \times 256$, followed by a random crop to $224 \times 224$ and random horizontal flipping. The size of the dynamic offline memory queue is set to match the cardinality of each respective dataset. For the evidential classification baselines, we maintain consistent optimizer and iteration settings to ensure a fair comparison. Finally, all retrieval metrics are reported for $K \in \{1, 5, 10\}$.

### .8. ECE Metric Calculation

To evaluate uncertainty reliability, we report Expected Calibration Error (ECE), calculated as the weighted average difference between empirical accuracy and predicted confidence across $M = 10$ bins:

$$\text{ECE} = \sum_{m=1}^{M} \frac{|B_m|}{N} \big| \text{acc}(B_m) - \text{conf}(B_m) \big|.$$

We adapt the definition of confidence $\hat{p}_i$ to the uncertainty mechanism. For analytic methods (Evidential, BTL), confidence is derived directly from the explicit uncertainty score as $\hat{p}_i = 1 - u_i$. For stochastic methods (Deep Ensembles, MC Dropout, PFE), we measure confidence via prediction consistency, defined as the fraction of latent samples or ensemble members agreeing with the majority prediction. This formulation ensures a consistent calibration assessment across both deterministic and probabilistic baselines.

