# OpenReview forum: "Evidential Retriever: Uncertainty-Aware Medical Image Retrieval"
_MIDL.io/2026/Conference — MIDL 2026 Poster_

### Official Review · Reviewer_9LGp · 2025-12-29

**Confidence:** 4
**Preliminary Rating:** 2
**Final Rating:** 4

**Summary:**

The paper introduces the Evidential Retriever, a dual-head Swin Transformer based architecture that jointly learns discriminative embeddings for content-based medical image retrieval and Dirichlet-based evidential uncertainty. The model is trained with a composite loss that couples a contrastive metric-learning objective and an evidential loss that predicts Dirichlet parameters. Experiments are performed on three datasets: ISIC 2017 dermoscopy, COVID-QU-Ex chest X-ray, and Kvasir endoscopy. The results are compared against deterministic, probabilistic, Bayesian, and evidential classification retrieval baselines.

**Strengths:**

The paper tackles an important gap in content-based medical image retrieval: current systems focus mainly on retrieval accuracy but provide little notion of how confident the model is about retrieved results. For clinical support tools, being able to flag uncertain or potentially spurious retrievals is highly valuable, and the paper motivates this need well in the introduction with references to CBMIR use-cases and uncertainty work. This gives the work clear practical relevance for the community.

**Weaknesses:**

The main weaknesses of the paper are the limited evaluation scope and the narrow set of baselines. The method is presented as a general uncertainty-aware retrieval framework, but the experiments only use three small datasets, which makes it hard to judge how well the approach would work in more realistic or diverse clinical settings. Medical retrieval problems often involve large, heterogeneous datasets with long-tailed pathology distributions, and testing on at least one such benchmark would give a clearer picture of how robust and broadly applicable the method actually is. In addition, the paper does not compare against any of the current foundation model baselines that are now widely used for medical retrieval.

**Detailed Comments:**

1. The current evaluation on ISIC, COVID-QU-Ex, and Kvasir spans three modalities (dermoscopy, CXR, endoscopy), which is a good starting point. However, because the paper aims to propose an “uncertainty-aware medical image retrieval” framework rather than a dataset-specific solution, I would strongly encourage the authors to broaden the experimental scope to at least one large-scale, clinically realistic retrieval benchmark.
For example, Denner et al. (2024) provide a unified CBMIR benchmark combining NIH14, CheXpert, MIMIC-CXR, and RadImageNet (~1.6M images across 12 anatomical regions and 185 classes). This benchmark captures long-tailed pathology distributions, heterogeneous imaging conditions, and substantial anatomical variety. Evaluating on even a subset of this benchmark would substantially strengthen the generalization and clinical relevance of the method.

2. Medical vision foundation models such as BiomedCLIP, MedCLIP, RAD-DINO, and DINOv2 have rapidly become the leading baselines for retrieval tasks, often outperforming classical metric-embedding approaches due to their broad semantic coverage and robust pretrained representations. To contextualize the contribution and ensure fair comparison, I suggest including additional foundation model baselines (e.g., frozen embeddings + kNN retrieval), and/or demonstrating that the evidential uncertainty head remains beneficial when attached to an encoder. This would clarify the method’s complementary role and show how it competes in foundation model driven retrieval.

3. The current OOD evaluation is based on only a single scenario: the model is trained on ISIC (ID) and uncertainty is tested on COVID-QU-Ex (OOD), which differs strongly in both content and imaging modality. While this setup shows that the proposed method can detect large distribution shifts, it does not reveal how well the uncertainty behaves under more clinically realistic shifts. In retrieval research, it is common to evaluate multiple ID/OOD combinations (within-modality shifts (e.g., ISIC → other dermoscopy datasets), within datasets shifts (based on metadata), cross-institution splits, or different acquisition settings). Including at least a few such additional OOD settings would give a more complete picture of robustness. One can take a look at T. J. Bungert, L. Kobelke, and P. F. Jaeger, “Understanding Silent Failures in Medical Image Classification".

4. Although the paper compares Swin Transformer and ViT backbones, the study remains strictly Transformer-only. This is a meaningful omission because CNN-based retrieval architectures (ResNet, VGG, and DenseNet) continue to serve as strong and computationally efficient baselines in medical CBIR, particularly for 2D datasets like ISIC, Kvasir, and chest X-ray tasks where local textures are important. Much of the classical CBMIR literature uses CNNs, so including at least one CNN comparator would provide a more historically grounded and comprehensive perspective on performance.
It is also unclear whether the evidential uncertainty head is intended to be general-purpose or inherently tied to Transformer features. If the method can operate on CNN embeddings, demonstrating this would strengthen the claim of architectural generality. If instead the approach relies on Transformer-specific properties (such as the CLS token or hierarchical self-attention) this should be explicitly stated. Either adding a CNN baseline or clarifying the architectural rationale would help readers understand the scope and applicability of the proposed method.

5. In Table 1, the Recall@[1,5,10] and mP@[1,5,10] metrics each contain three separate values, and different methods achieve the best score for different K-values. However, none of these best-performing values are bolded, even when a baseline clearly outperforms the proposed method. At the same time, the authors do bold results in other columns (mAP, AUROC, AUPRC, ECE ) where their model achieves the best score. This inconsistent use of boldface can unintentionally give the impression that the proposed method is the strongest across most metrics simply because the highlighted numbers appear only in its favor.
Suggested: Bold the best value for each K independently, regardless of which method achieves it. While this may make the table visually a bit busier, it provides a fair and transparent comparison. If the authors still wish to draw attention to their own method, they could use a separate, non-ranking visual editing.

**Justification Of Final Rating:**

I appreciate the authors thorough and thoughtful responses to the review comments. The addition of large-scale evaluations on CheXpert (224k images) and NIH-CXR14 (112k images) substantially strengthens the empirical scope of the paper and better supports the claim that the proposed method scales beyond small curated datasets. The newly added comparisons with foundation models (BiomedCLIP, DINOv2, RAD-DINO) play an important role in contextualizing the contribution, helping clarify how SSL-pretrained encoders behave in retrieval settings.

Furthermore, the inclusion of a within-modality OOD experiment (ISIC → PAD-UFES-20) provides a more clinically realistic assessment of uncertainty under subtle distribution shifts, strengthening the paper’s robustness claims.

With these revisions, the paper now offers relevant insights into uncertainty-aware medical image retrieval and provides useful empirical evidence on how evidential learning interacts with both domain-specific models and large foundation models.

Although the rebuttal significantly strengthens the paper, some limitations remain. While CheXpert and NIH-CXR14 provide valuable large-scale validation, both datasets are restricted to chest X-rays and still under 1M images. Extending the analysis to additional large-scale datasets from different imaging modalities (e.g., CT or MRI) would further support the generality of the approach. Similarly although one within-modality OOD experiment is now included, the robustness evaluation is still limited to a small number of shift types. Additional OOD scenarios like cross-institution shifts, protocol changes or rare pathology subsets, would provide a more comprehensive picture of uncertainty behavior in realistic clinical settings. These aspects represent promising directions for future work.

In conclusion, the updated version presents contributions that are valuable to the community, motivating a weak accept.

**Justification Of The Preliminary Rating:**

While the paper presents a well motivated approach to uncertainty-aware medical image retrieval and offers a clean evidential formulation, the experimental evidence is not yet sufficient to support the strength or generality of the claims. The method is positioned as a broadly applicable retrieval framework, yet the evaluation is limited to three relatively small datasets and only a single, modality-mismatched OOD scenario (ISIC to COVID-QU-Ex). As a result, it is difficult to assess how the proposed uncertainty behaves under more realistic distribution shifts or in settings with the long-tailed, heterogeneous data distributions typically encountered in clinical applications.

Also the comparison set omits several important baselines. No CNN-based retrieval backbone is included, despite CNNs being widely used and often strong performers on 2D medical imaging tasks. More importantly, the paper does not compare against modern foundation model encoders, which currently represent the standard in medical image retrieval and would provide a stronger point of reference for evaluating the proposed method’s contribution. Without these baselines, it is challenging to contextualize the performance improvements or determine whether the evidential head offers advantages beyond already established representation learning techniques.

The core idea is interesting and has potential value, but the current level of validation does not fully support the claims. The limited OOD/ID evidence and absence of key baselines make it difficult to assess how broadly applicable or competitive the proposed approach truly is. For these reasons, a weak reject seems appropriate at this stage. I hope the authors can incorporate the missing comparisons and expand the evaluation in the rebuttal, as doing so would substantially strengthen the paper and clarify the practical impact of the proposed method.

**Questions To Address In The Rebuttal:**

1. Can the authors clarify why the evaluation is limited to three small datasets, and whether they can provide additional retrieval or uncertainty results on a larger and more diverse benchmark?

2. Can the authors justify the absence of foundation model baselines (e.g., BiomedCLIP, MedCLIP, RAD-DINO, DINOv2) and explain how the proposed method is expected to compare against these stronger encoders? If feasible, providing even one FM baseline in the rebuttal would significantly strengthen the paper.

3. Could the authors comment on whether the evidential head is intended to be architecture agnostic?

4. Can the authors provide more realistic OOD/ID evidence (for example, results from within-modality shifts)?

---

> ### Author Response · Authors · 2026-01-24
>
> We thank the reviewer for the critical and constructive feedback. Your suggestions to include Foundation Models and larger datasets have significantly strengthened the paper. We have extensively revised the manuscript to address all the points raised.
>
> Q1: Can the authors clarify why the evaluation is limited to small datasets and provide additional retrieval or uncertainty results on a larger and more diverse benchmark?
>
> To address the concern about scale, we have added Section 4.1 and Table 2, evaluating our method on two large-scale benchmarks: CheXpert (224k images) and NIH-CXR14 (112k images). The results demonstrate three key findings:
>
> 1) Evidential Upgrade over Frozen FMs: Training our evidential heads on top of frozen Foundation Models (e.g., RAD-DINO + 2 Heads) consistently outperforms the "off-the-shelf" frozen baselines. For instance, on CheXpert, the evidential upgrade boosts RAD-DINO's mAP from 38.25% to 46.47% while providing calibration (ECE 0.1028), confirming the benefit of our evidential formulation.
>
> 2) Superiority over Evidential Classification: Our Evidential Retriever outperforms the  Evi classification baseline in accuracy (e.g., 47.37% vs. 46.66% mAP on CheXpert) and calibration. Specifically, the RAD-DINO backbone upgraded with our heads achieved the best calibration across both datasets, confirming our framework provides more reliable uncertainty estimates than the baseline, demonstrating the value of our unified metric-learning objective.
>
> 3) Competitive Performance at Scale: Our lightweight Swin-Small model achieves 47.37% mAP on CheXpert, surpassing even the massive frozen FMs like DINOv2 (36.73%) and BiomedCLIP (41.59%). Notably, equipping the RAD-DINO backbone with our evidential heads yields the highest NIH performance (24.66% mAP), while our lightweight Swin-Small model remains highly competitive (24.37% mAP).
>
> Q2: Can the authors justify the absence of foundation model baselines and explain how the method compares?
>
> To address the absence of Foundation Models, we have added a comprehensive comparison against BiomedCLIP, DINOv2, and RAD-DINO in Table 2 (CheXpert/NIH) and Appendix Table 4 (ISIC, COVID-QU-Ex, Kvasir). We found that "off-the-shelf" frozen FMs struggle with calibration and domain shifts-for example, frozen RAD-DINO achieves only 27.21% mAP on Kvasir. However, we demonstrate that our Evidential Head acts as a powerful architecture-agnostic "safety upgrade." As noted in Response for Q1, training our heads on top of frozen RAD-DINO boosts performance on large-scale datasets while enabling essential calibration (ECE 0.1028); similarly, on specialized datasets like COVID-QU-Ex, this upgrade yields massive gains, proving our method effectively complements stronger encoders.
>
> Crucially, our specialized Evidential Retriever (Swin-Small) consistently achieves competitive or superior performance, particularly on specialized datasets. As shown in Appendix Table 4, our model outperforms the "upgraded" FMs on domain-specific tasks; for instance, on Kvasir, our model achieves 91.99% mAP, far exceeding the RAD-DINO + 2 Heads result of 65.15%. While the upgraded RAD-DINO performs slightly better on NIH-14 (24.66% vs. 24.37%), our Swin-Small model remains highly effective across the board, confirming that a domain-specific architecture trained with our metric-evidential objective offers a robust alternative to large-scale FMs. Accordingly, we have updated Section 2 (Related Work) to discuss these recent foundation model approaches.
>
> Q3: Can the authors provide more realistic OOD evidence (e.g., within-modality shifts)?
>
> We expanded our OOD evaluation in Section 4 (“Robustness to Within-Modality Domain Shift”) and Figure 3 to include a realistic "Within-Modality" shift scenario. We trained the model on ISIC and tested it on PAD-UFES-20 (smartphone-acquired lesions). Unlike the gross cross-modality shift (ISIC $\to$ COVID), this mimics the subtle domain shift encountered in real-world tele-dermatology. Our model achieves an AUROC of 0.7261 in this setting, outperforming the evidential classification baseline (0.6909) and demonstrating superior sensitivity to fine-grained acquisition artifacts.
>
> Q4: Could the authors comment on whether the evidential head is architecture agnostic and include CNN comparisons?
>
> We have now included a detailed comparison of CNN backbones (ResNet-50, DenseNet-121) against Vision Transformers (Swin, ViT). This study confirms that our evidential head is architecture-agnostic and can be effectively applied to CNNs. However, while CNNs are competitive in accuracy, Swin-Small consistently yields superior calibration (ECE $\approx$ 0.05 vs. $>0.20$ for CNNs). Consequently, Swin-Small emerges as the best-performing backbone overall, offering the optimal balance of high accuracy and reliable calibration. This analysis is detailed in Appendix .6.
>
> Also, Table 1 is now corrected to bold the best value for each $K$ independently, ensuring fair visual comparison.

---

> ### Author Response · Authors · 2026-01-30
>
> We thank the reviewer for appreciating the additional work and for raising the score. We are grateful for the suggested future directions. Given the current computational constraints, we view these as important next steps and will pursue them in future work.

---

### Official Review · Reviewer_DP41 · 2026-01-08

**Confidence:** 3
**Preliminary Rating:** 4
**Final Rating:** 5

**Summary:**

This paper proposes the Evidential Retriever which is trained to retrieve clinically relevant and visually similar images from a data source given a query image while providing a confidence measure to ensure reliability. This framework simultaneously learns embeddings for retrieval and parameters of a Dirichlet distribution for evidential deep learning via a new loss function.

**Strengths:**

- The authors address a clinically relevant gap
- Evaluation on multiple datasets with different modalities
- Meaningful baselines
- Extensive ablation studies of loss components and backbone choice
- Evaluation of uncertainty quantification: calibration via Expected Calibration Error, sparsification curves, OOD classification and qualitative analyses
- Informative figure of architecture (Figure 1)

**Weaknesses:**

- Section 3.1: The introduction into the class probabilities ($p_k$) and uncertainty ($u$) could be extended to make it clearer
- The qualitative analyses in the appendix (.1, .2, .3, .4,) are as meaningful as the OOD evaluations; for space reasons potentially move the backbone ablation study to the appendix
- Code not available

**Detailed Comments:**

- The first paragraph in the related work section gives a bit of a jumpy introduction into the topic. It seems surprising to immediately start reading about different loss functions.
- “The closest work (Dordevic and Kumar, 2024) relies on evidential classification foundations, i.e. they use classification-trained CLS token as an embedding. One of their other proposed approach is a Naive strategy to incorporate metric learning would require two networks - one for retrieval and one for uncertainty.“ unclear syntax
- Minor comment: section 3.1 first paragraph (p.3 bottom) repetition of “specifically”

**Justification Of Final Rating:**

The authors have addressed all of my issues to my satisfaction during the rebuttal. The paper addresses a clinically meaningful gap and is supported by rigorous experimental evidence showing that uncertainty quantification indeed improves image retrieval.

**Justification Of The Preliminary Rating:**

Overall, the paper proposes an interesting idea. Adding uncertainty quantification is not only clinically relevant but it seems to even improve the embedding space (see Appendix .1), certain weaknesses remain.

**Questions To Address In The Rebuttal:**

- Address the points raised in the detailed comments
- Please make the introduction into Dirichlet parameters and uncertainty metric more extensive

---

> ### Author Response · Authors · 2026-01-24
>
> We thank the reviewer for the detailed feedback and for recognizing the clinical relevance of our uncertainty quantification approach. We have incorporated all your suggestions into the revised manuscript, highlighting these changes.
>
> Q1: Can the authors address the syntax issues and smooth the introduction in the related work section?
>
> We have rewritten the first paragraph of the Related Work section to provide a smoother and more logical transition from general CBMIR goals to deep metric learning. Additionally, we have corrected the syntax regarding the "closest work" paragraph to ensure clarity and improve flow, as requested.
>
> Q2: Can the authors make the introduction into Dirichlet parameters and uncertainty metric more extensive? [Section 3.1]
>
> We have clarified the definitions in Section 3.1. We now explicitly define epistemic uncertainty as $u = K/S$, explaining that it is inversely proportional to the total evidence $S$. We further clarified the relationship between evidence and uncertainty: high evidence accumulation ($S$) results in low uncertainty ($u \to 0$), while OOD or ambiguous samples with scarce evidence yield high uncertainty ($u \to 1$).
>
> Q3: Can the authors move the backbone ablation study to the appendix for space reasons?
>
> We agree with this suggestion. We have moved the detailed Backbone Ablation Study (comparing CNNs, Swin Transformers, ViT) to the Appendix( section .6) . To ensure the main text remains self-contained, we added a summary sentence in Section 4.3 referencing this appendix and noting that the Swin-Small backbone was selected because it consistently provided the best trade-off between retrieval accuracy and calibration across the datasets.
>
> Q4: Is the code available?
>
> We will make the code publicly available upon acceptance to ensure reproducibility.

---

> > ### Comment · Reviewer_DP41 · 2026-01-29
> >
> > Thank you for addressing all mentioned issues. I really appreciate the revision of Sec. 3.1. Good luck with your submission!

---

### Official Review · Reviewer_Dbzr · 2026-01-12

**Confidence:** 4
**Preliminary Rating:** 4
**Final Rating:** 5

**Summary:**

In this work, authors propose an architecture called "Evidential Retriever" for a content-based medical image retrieval system. Evidential Retriever architecture involves combining a representation learning model with evidential uncertainty modeling. The decision of retrieval is then made by a combination of using the similarity and uncertainty metrics.

**Strengths:**

The authors present a well-structured and carefully executed study that is supported by a clear and thoughtful experimental design. Each component of the methodology is logically motivated, making it straightforward to understand how the proposed approach was developed, evaluated, and validated.

**Weaknesses:**

While the work is supported by strong experimental design and scientific rigor, the chosen data and visualizations do not clearly demonstrate the method’s behavior in realistic clinical settings. In particular, it is unclear how the approach would perform on heterogeneous clinical datasets that vary across modalities, comorbidities, demographics, and acquisition protocols. The use of relatively controlled data is appropriate for validating the core methodology, but it limits insight into robustness and generalizability. Similarly, the visualizations emphasize aggregate trends and may obscure variability across clinically meaningful subgroups. Including experiments or stratified analyses that reflect real-world clinical diversity would strengthen the paper and better establish its relevance to practical clinical applications.

**Detailed Comments:**

No further comments.

**Justification Of Final Rating:**

The authors have satisfactorily addressed all reviewer comments with clear explanations and appropriate revisions. As a result, the concerns raised earlier have been resolved, and I have accordingly updated my final rating to reflect these improvements.

**Justification Of The Preliminary Rating:**

The authors present a method that is grounded in strong experimental design and scientific rigor, with results that clearly validate the core technical contribution under controlled conditions. The questions around large-scale performance and high-dimensional heterogeneity are natural next steps rather than fundamental shortcomings of the approach. The clarity of the methodology and the transparency of the experimental evaluation provide confidence that such extensions are feasible.

**Questions To Address In The Rebuttal:**

Can the authors comment on computational feasibility, memory requirements, and whether performance trends observed in the current experiments are expected to hold at this scale?

How robust is the method to the substantial variability present in real-world clinical datasets, where heterogeneity may arise simultaneously from modality differences, comorbidities, demographics, and acquisition protocols?

---

> ### Author Response · Authors · 2026-01-24
>
> We thank the reviewer for the positive assessment and for acknowledging the scientific rigor of our experimental design. We have significantly expanded our evaluation to address your questions regarding scalability and heterogeneity. The revised manuscript includes these expanded evaluations, with all changes highlighted in red.
>
> Q1: Can the authors comment on whether performance trends observed in the current experiments are expected to hold at scale on heterogeneous datasets?
>
> We expanded our evaluation to CheXpert (224k images) and NIH-CXR14 (112k images) in Section 4.1. Results (Table 2) confirm that our performance holds effectively at scale. On CheXpert, our Swin-Small model (47.37% mAP) outperforms frozen FMs like DINOv2 (36.73%) and BiomedCLIP (41.59%) while maintaining superior calibration. A similar trend is observed on NIH-14, where our method (24.37% mAP) consistently surpasses frozen baselines. Notably, equipping the RAD-DINO backbone with our evidential heads yields the highest NIH performance (24.66% mAP), while our model remains highly competitive (24.37% mAP). This validates that our framework is both effective as a standalone model and as a modular "safety upgrade" for pre-trained encoders.
>
> Q2: How robust is the method to the substantial variability present in real-world clinical datasets?
>
> To evaluate robustness against realistic clinical shifts, we expanded our OOD evaluation by adding a "Within-Modality" experiment (Section 4  - “Robustness to Within-Modality Domain Shift” para) and Figure 3 - training on ISIC (standardized dermoscopy) and testing on PAD-UFES-20 (smartphone images). This mimics the shift to consumer-grade devices in tele-dermatology. Our model achieves an AUROC of 0.7261 in detecting these OOD samples, outperforming the evidential classification baseline (0.6909). This demonstrates that our unified loss is sensitive to fine-grained acquisition shifts, offering improved reliability for heterogeneous environments.
>
> Q3: Can the authors comment on computational feasibility and memory requirements?
>
> Our primary efficiency advantage is that the Evidential Retriever is a single-pass uncertainty estimator. Unlike Deep Ensembles or MC Dropout, which require multiple forward passes (typically 5–10) to estimate uncertainty, our model produces both embeddings and uncertainty in a single pass, ensuring low latency. Additionally, our Swin-Small backbone (~50M params) is significantly lighter than many Foundation Model encoders, striking a favorable balance between performance and deployment constraints in resource-limited settings.

---

> ### Comment · Area_Chair_DSyP · 2026-02-01
>
> Hi reviewer Dbzr,
>
> Thank you for your hard work so far. Could you please check the author response and update your final rating?
>
>  - by clicking “Edit” → “Official Review” and providing the Final Rating by February 1st 2026 (23:59 AoE).
>
> Thanks,
>
> Your AC

---

### Author Rebuttal · Authors · 2026-01-24

**Rebuttal:**

We thank the Reviewers for their constructive feedback and for recognizing the potential of our work. We were encouraged that the reviewers found our experimental design to be supported by "scientific rigor" (Reviewer Dbzr), recognized that we address a "clinically relevant gap" (Reviewer DP41), and acknowledged the "practical relevance" of uncertainty-aware retrieval for clinical support tools (Reviewer 9LGp).

In this work, we introduced the Evidential Retriever, a unified framework that seamlessly integrates evidential uncertainty quantification into deep metric learning. Our goal is to move beyond deterministic retrieval to systems that are both accurate and calibrated.

In response to the valuable feedback, we have carefully revised the manuscript to address all the points raised and have incorporated the reviewer's suggestions to strengthen the work (changes highlighted in red). We have expanded our experimental validation to include:

1) Large-Scale Benchmarking [Sec 4.1]: We extended our evaluation to CheXpert and NIH-14 (over 330k images combined), demonstrating that our method scales effectively to heterogeneous, long-tailed clinical distributions.
2) Foundation Model Comparison [Sec 4.1 & Appendix Sec .5)]: We now compare against and integrate with state-of-the-art Foundation Models (BiomedCLIP, DINOv2, RAD-DINO). We demonstrate that our framework acts as an architecture-agnostic "safety upgrade," significantly improving the calibration of these powerful encoders.
3) Realistic OOD Testing [Sec 4]: We added a "within-modality" shift experiment (ISIC $\to$ PAD-UFES-20), confirming our model's robustness to real-world acquisition artifacts.
4) Backbone Ablation (CNNs vs Swin Transformers vs ViT) [Appendix (Sec .6)]: We added a comprehensive comparison against CNN baselines (ResNet, DenseNet), confirming that while CNNs are competitive in accuracy, our Swin Transformer backbone offers superior performance and uncertainty calibration.

Furthermore, based on reviewer suggestions, we improved the manuscript's structure and clarity. We moved the Qualitative Analysis of COVID_QU_Ex to the main text to better highlight our model's interpretability and shifted the Backbone Ablation to the Appendix, and clarified key theoretical concepts (e.g., the Dirichlet formulation).

We provide detailed individual responses below addressing specific queries. We thank the reviewers again for their time and insights, which have substantially strengthened this work.

**Supporting Material:**

/attachment/249daed4ed668aa957079d4136fd5edd02e3d8fd.pdf

---

### Meta-Review · Area_Chair_DSyP · 2026-02-09

**Recommendation:** Accept (Poster)
**Confidence:** 5

**Metareview:**

Consistent positive reviews lead to a clear acceptance. No oral recommendation.

---

### Decision · Program_Chairs · 2026-02-13

Accept (Poster)